# Machine Learning Approach for Predicting Hypertension Based on Body Composition in South Korean Adults

**DOI:** 10.3390/bioengineering11090921

**Published:** 2024-09-14

**Authors:** Jeong-Woo Seo, Sanghun Lee, Mi Hong Yim

**Affiliations:** 1Digital Health Research Division, Korea Institute of Oriental Medicine, Daejeon 34504, Republic of Korea; jwseo02@kiom.re.kr; 2KM Data Division, Korea Institute of Oriental Medicine, Daejeon 34504, Republic of Korea; ezhani@kiom.re.kr

**Keywords:** hypertension, machine learning, body composition, body fat mass, lean mass

## Abstract

(1) Background: Various machine learning techniques were used to predict hypertension in Korean adults aged 20 and above, using a range of body composition indicators. Muscle and fat components of body composition are closely related to hypertension. The aim was to identify which body composition indicators are significant predictors of hypertension for each gender; (2) Methods: A model was developed to classify hypertension using six different machine learning techniques, utilizing age, BMI, and body composition indicators such as body fat mass, lean mass, and body water of 2906 Korean men and women; (3) Results: The elastic-net technique demonstrated the highest classification accuracy. In the hypertension prediction model, the most important variables for men were age, skeletal muscle mass (SMM), and body fat mass (BFM), in that order. For women, the significant variables were age and BFM. However, there was no difference between soft lean mass and SMM; (4) Conclusions: Hypertension affects not only BFM but also SMM in men, whereas in women, BFM has a stronger effect than SMM.

## 1. Introduction

South Korea is currently facing serious issues of low birth rates and an aging population. As the population of newborns decreases, the average age of society members rises, emphasizing the importance of health management for those actively participating in economic activities [1]. Consequently, there is a need for managing various chronic adult diseases. However, access to clinical assistance has limitations in terms of accessibility and cost, highlighting the need for methods that can be easily managed at home [2]. Common methods that can be relatively easily monitored and controlled in daily life include weight and blood pressure. More specifically, body composition measurement devices have been released, allowing for more detailed analysis of the composition of different parts of the body, thus facilitating easier health management.

Body composition refers to the components that make up the human body, serving as an indicator of individual health characteristics through an expanded interpretation at the molecular level [3]. The molecular components of body composition include body fat, body water, protein, and minerals. Various body composition analyzers are used to measure these components. For body composition analysis, a small constant current of microamperes is sent through the body, and the voltage is measured to calculate the impedance, which is then analyzed [4]. Compared to the body mass index (BMI), which is the ratio of weight to height, body composition analysis is known to have higher reliability and validity as it provides a more detailed breakdown of body components [5]. There are various indicators for body composition analysis, including skeletal muscle mass, fat mass, muscle mass by body part, and body water, which are known to be indicators that can identify the presence of various diseases [6]. In particular, fat mass and muscle mass have been shown in various prior studies to be potential predictors of metabolic syndrome diseases such as hypertension, as well as cardiovascular diseases like arteriosclerosis [7,8]. Among these metabolic and cardiovascular diseases, a key clinical risk factor is hypertension. Hypertension is a straightforward indicator that can be checked using a simple blood pressure monitor; however, its measurement is limited to the systolic and diastolic values and it does not provide insights into the internal causes and conditions of the body that reflect these blood pressure values [9]. Understanding how the conditions of fat or muscle at specific body locations influence these values requires additional statistical methods and clinical interpretations to prove causality. Therefore, to manage such cardiovascular diseases, it is necessary to design or plan clinical approaches and personal health management methods based on the correlation and connectivity of more detailed elements and hypertension. Various prior studies have been conducted to develop predictive models based on the correlation between body composition indicators and hypertension for this purpose. In a study that examined the associations of body composition with blood pressure and hypertension using indicators from dual-energy X-ray absorptiometry of body composition and total skeletal muscle, body fat percentage and fat distribution were suggested as effective features for the model in predicting hypertension [10]. Another prior study examined the correlation between body size and composition and elevated blood pressure in adolescent girls, confirming the association between fat-free mass and blood pressure [11]. Another study developed a machine learning-based model to predict hypertension using body composition features without distinguishing between genders. In this study, body fat mass was included as a feature in the model, which achieved up to 90% prediction accuracy [12]. These prior studies commonly identified fat mass among body composition elements as a common indicator for classifying or predicting hypertension.

In addition to common indicators, various variables are presented as factors of hypertension but the results of most hypertension prediction models vary depending on the gender and demographic characteristics of the subjects. In addition, it is necessary to develop a prediction model using indicators that affect the occurrence of hypertension. Therefore, in order to identify hypertension indicators that take into account the characteristics of Koreans, priority should be given to identifying variables that differ in each gender.

This study aims to develop prediction models that can classify the presence of hypertension in Korean adults aged twenty and above, using various machine learning techniques and distinguishing between the models for men and women. This study also aims to verify the importance of the features included in the models.

## 2. Materials and Methods

### 2.1. Data Sources and Participant Selection

In this study, data collected through cross-sectional research were used from April 2022 to December 2022 at five hospitals in the Republic of Korea, including Dongguk University Ilsan Oriental Hospital, Pusan National University Korean Medicine Hospital, Gachon University Gill Hospital, Dongshin University Korean Medicine Hospital, and Semyung University Korean Medicine Hospital. This study was approved by the Institutional Review Boards of each hospital and was conducted according to the Declaration of Helsinki (IRB No., Dongguk University Ilsan Oriental Hospital, DUIOH-2022-01-005; Pusan National University Korean Medicine Hospital, PNUKHIRB-2022-02-001; Gachon University Gill Hospital, GIRB-22-101; Dongshin University Korean Medicine Hospital, NJ-IRB-013; Semyung University Korean Medicine Hospital, SMJOH-2022-06). All participants signed an informed consent form prior to enrollment. 

A total of 3000 participants were assessed for eligibility for inclusion in the study. Of the 3000 participants, 2986 completed the clinical study after excluding individuals who did not meet inclusion/exclusion criteria. The inclusion criteria were as follows: (1) adults aged 19 years or older; (2) those without cognitive disorder; (3) those who voluntarily decided to participate after fully understanding the study, and agreed to participate in the study via written consent. The exclusion criteria were as follows: (1) those who were unable to measure their health status with a device due to mobility difficulties; (2) those who were diagnosed by a doctor with diseases such as cardiovascular diseases, cerebrovascular diseases, malignant neoplasms, mental diseases, arthritis, or thyroid disease, and have been suffering from them since diagnosis; (3) women who were pregnant, might be pregnant, or were menstruating; (4) those without a mobile phone in their name; (5) persons judged by the principal investigator to be unsuitable for participation. Among 2986 participants who completed the study, 80 participants with errors in the measurements of body composition were excluded. Finally, 820 men and 2086 women were included in the analysis (Figure 1).

The hypertension group was defined by two questions: “Have you ever been diagnosed with hypertension by a physician?” and “Have you been suffering from hypertension since diagnosis”. Participants who answered “Yes” to both questions were classified into the hypertension group, and those who did not answer “Yes” to both questions were classified into the non-hypertension group.

### 2.2. Body Composition Measurement

Body composition indicators were calculated after measurement by a well-trained clinical research coordinator according to standardized protocols using a multifrequency bioelectric impedance analyzer (BWA 2.0; In-Body Co., Seoul, Republic of Korea). Subjects were asked to wear light clothes and to have a minimum fasting period of four hours for the measurement. The measurement was conducted after waiting for five minutes with the subject lying on the bed and wearing clamp-type sensors on both wrists and ankles. Individual variables for body fat mass, lean mass, and body water were obtained (Table 1). 

### 2.3. Model Development and Assessment

All statistical analyses were implemented using R version 4.2.1 (R Foundation for Statistical Computing, Vienna, Austria) [13]. In all statistical tests, two-tailed tests were applied and a significance level of 0.05 was used. To take sex-specific differences in body composition indicators into account, all statistical analyses were performed by sex. Data were randomly split into training and test sets in a 7:3 ratio. The training set was used for training and validation of model development and the test set was used for final model evaluation. To compare general characteristics and body composition indicators between the training set and test set, independent two-sample t-tests and chi-square tests were used for continuous and categorical variables in each sex, respectively. To compare general characteristics and body composition indicators between hypertension and non-hypertension groups, independent two-sample t-tests were used in the training set by sex. In each sex, simple logistic regression analyses were used in the training set to identify the association of individual body composition indices with hypertension.

To build the models, previously developed machine learning algorithms were applied using the Caret R package [14,15,16]. The Caret package has several functions, including data splitting, data preprocessing, feature selection, automatic hyperparameter tuning, training, variable importance estimation, and model assessment, which can streamline the processes of model development and evaluation.

The hypertension classification models using the combined body composition indicators were developed in the training set through six machine learning algorithms: elastic-net-regularized logistic regressions (E-net), k-nearest neighbor (K-NN), random forest (RF), support vector machine (SVM), extreme gradient boosting (XGBoost), and neural network (NN). E-net is a penalized linear or logistic regression method that combines ridge and lasso regressions, and has functions of efficient shrinkage regularization and feature selection [17]. Additionally, E-net is an effective regression method that secures the stability and generality of the model by reducing the impact of multicollinearity when multiple highly correlated explanatory variables are included. K-NN is a non-parametric method for classification and regression that works based on the similarity of data points in a given dataset [18]. K-NN is a simple and easy-to-implement method. RF is an ensemble machine learning method for classification, regression, and unsupervised learning based on a set of multiple trees [19]. RF is known to be robust to data with missing values, handles large and complex data, and reduces overfitting. SVN is a machine learning algorithm for linear or non-linear classification and regression that solves classification problems by constructing hyperplanes with optimal separation and maximum margin between individual classes [20]. SVM is known to be effective for high-dimensional data and can handle complex or non-linear relationships using kernel functions. XGBoost is a decision-tree-based ensemble machine learning method for classification and regression [21]. XGBoost can control model complexity and overfitting by adding a regular term in the loss function. NN is a machine learning algorithm that works similarly to the neural pathways in the human brain [22]. NN is composed of three connected layers of input, hidden and output, and is known to be powerful in capturing complex patterns and relationships in complex data. The hyperparameters were tuned to obtain the maximum area under the receiver operating characteristic curve (AUROC) using 5-fold stratified cross-validation in the training set to be randomly divided into the training and validation subsets in a 4:1 ratio. To investigate the contribution of body composition variables selected in the models, the relative importance of the variables was calculated.

The performance of each model was evaluated by constructing a receiver operating characteristic (ROC) curve and a confusion matrix for binary classification in the test set [23,24,25]; AUROC was obtained from the ROC curve, and kappa, F1 score, precision, accuracy, sensitivity, and specificity were obtained from the confusion matrix. The confusion matrix depends on the chosen threshold value. To obtain the optimal threshold value for each model, Youden’s index was used. AUROC is obtained by calculating the area under the ROC curve. The ROC curve is a curve drawn according to different threshold settings from 0 to 1, with the vertical axis representing sensitivity and the horizontal axis representing 1-specificity. To compare the AUROC of the model with the best AUROC value to the AUROC of each of the other models, *p* values were derived using the z score [26,27]. The accuracy is calculated as the proportion of accurately predicted samples to the total number of samples. Sensitivity and specificity refer to the proportion of predicted positive samples to the total number of observed positive samples and the proportion of predicted negative samples to the total number of observed negative samples, respectively. Precision presents the proportion of observed positive samples to the total number of predicted positive samples. F1 score is a combined measure of precision and sensitivity, calculated as the harmonic mean of precision and sensitivity. Kappa indicates the degree of agreement between the predicted and observed outcomes, and is calculated taking into account coincidence. The measures of performance evaluation were presented along with the 95% confidence interval (CI) obtained from 2000 bootstrap repetitions.

## 3. Results

### 3.1. General Characteristics and Body Composition

The data were randomly divided into the training and test sets in a 7:3 ratio by sex. Of a total of 820 men, 575 were included in the training set and 245 were included in the test set. Among a total of 2086 women, 1462 were included in the training set and 624 were included in the test set. In both men and women, there were no significant differences between the training and test sets in the percentage of participants in the hypertension group, or general characteristics such as age, temperature, blood pressure, pulse rate, height, weight, BMI, and other body composition variables (Table 2).

Of a total of 575 men in the training set, 75 were included in the hypertension group and 500 were included in the non-hypertension group. Among the 1462 women in the training set, 108 were included in the hypertension group and 1354 were included in the non-hypertension group. For men, there were significant differences between the hypertension and non-hypertension groups in age, blood pressure, height, and BMI in the training set. For women, there were significant differences between the hypertension and non-hypertension groups in age, blood pressure, height, weight, and BMI in the training set. For both men and women, significant differences were found between the hypertension and non-hypertension groups in several variables related to body fat mass, lean mass, and body water as well as additional variables such as visceral fat level, visceral fat area, proportion of weight to ideal weight, proportion of total body water to fat-free mass, and the 50 kHz phase angle of the trunk in the training set (Table 3).

### 3.2. Association of Individual Body Composition Variables with Hypertension

To investigate the association between individual body composition variables and hypertension, crude analyses were conducted in the training set (Table 4). Older men and women were more likely to be classified into the hypertension group (odds ratio [95% CI]; men, 4.32 [3.09, 6.06]; women, 3.36 [2.57, 4.4]). Men and women with higher BMI were also more likely to be in the hypertension group (men, 1.5 [1.2, 1.89]; women, 1.82 [1.55, 2.14]). For both men and women, hypertension was positively associated with most body fat mass variables. Men and women with higher percentages of body fat mass to weight were more likely to be in the hypertension group (men, 2.7 [2.02, 3.61]; women, 2.2 [1.79, 2.71]). In men, hypertension was negatively associated with lean mass variables of soft lean mass and skeletal muscle mass, whereas in women, hypertension had no association with these variables. Men and women with higher percentages of skeletal muscle mass to weight were less likely to be in the hypertension group (men, 0.33 [0.24, 0.44]; women, 0.44 [0.35, 0.54]). In men, hypertension was negatively associated with total body water, intracellular water, and extracellular water, whereas in women, hypertension was not associated with any of those variables. Men and women with higher percentages of total body water to weight were less likely to belong to the hypertension group (men, 0.38 [0.29, 0.51]; women, 0.46 [0.37, 0.57]). Men and women with higher proportions of extracellular water to total body water were more likely to belong to the hypertension group (men, 2.08 [1.61, 2.69]; women, 1.62 [1.33, 1.99]). In both men and women, hypertension was positively associated with visceral fat level, visceral fat area, proportion of weight to ideal weight, and proportion of total body water to fat-free mass. Men and women with a higher 50 kHz phase angle of the trunk were less likely to be in the hypertension group (men, 0.36 [0.27, 0.49]; women, 0.69 [0.56, 0.84]).

### 3.3. Comparison of Performance of the Models

The ROC curve was drawn and the AUROC value was calculated for each model in the test set (Figure 2, Table 5). AUROC values of the six models ranged from 0.783 to 0.865 for men and 0.774 to 0.831 for women. For both men and women, the E-net model reported the highest AUROC (AUROC [95% CI]; men, 0.865 [0.806, 0.914]; women, 0.831 [0.78, 0.881]). This was followed by the NN model with an AUC of 0.853 [0.797, 0.904] and the XGBoost model with an AUC of 0.827 [0.761, 0.883] for men, and was followed by the XGBoost model with an AUC of 0.826 [0.772, 0.876] and the NN model with an AUC of 0.822 [0.771, 0.871] for women. For both men and women, the SVM model has the lowest AUROC (men, 0.783 [0.697, 0.857]; women, 0.774 [0.712, 0.831]). Comparing the AUROC of the model with the best AUROC value to the AUROC of each of the other models, there were significant differences in AUROC between the E-net model and the RF model and between the E-net model and the SVM model in men, while there were no significant differences in AUROC between the E-net model and each of the other models in women.

Kappa, F1 score, precision, accuracy, sensitivity, and specificity were calculated using the classification confusion matrix obtained based on the optimal threshold for each model (Table 5). In men, the NN model exhibited the best kappa, F1 score, precision, and accuracy and achieved a well-balanced sensitivity and specificity (kappa, 0.365 [0.257, 0.469]; F1 score, 0.489 [0.37, 0.586]; precision, 0.33 [0.232, 0.424]; accuracy 0.743 [0.69, 0.796], sensitivity 0.941 [0.833, 1], specificity 0.714 [0.653, 0.771]). This was followed by the E-net model and then the XGBoost model. In women, the XGBoost model had the highest kappa, F1 score, precision, and accuracy and a good balance between sensitivity and specificity (kappa, 0.265 [0.186, 0.345]; F1 score, 0.347 [0.262, 0.43]; precision, 0.223 [0.162, 0.291]; accuracy 0.79 [0.758, 0.821], sensitivity 0.78 [0.653, 0.896], specificity 0.791 [0.758, 0.824]). The XGBoost model was followed by the NN model and then the E-net model.

As a result of the model evaluation, the E-net model showed the best performance on most evaluation metrics for both men and women, followed by the XGBoost model and the NN model. For each final model, a selected subset of variables and their importance were obtained (Figure 3). In the E-net model, three variables were selected for men: age; body fat mass percentage of trunk; and percentage of total body water to weight. In the E-net model, nine variables were selected for women: age; visceral fat level; proportion of total body water to fat-free mass; BMI; proportion of weight to ideal weight; body fat mass of trunk; body fat mass percentage of the left leg; percentage of body fat mass to weight; and percentage of total body water to weight. In the XGBoost model and the NN model, age and the 50 kHz phase angle of the trunk were selected with relatively high importance for men, and age and proportion of total body water to fat-free mass were selected with relatively high importance for women.

## 4. Discussion

In this study, we examined the correlations and differences in various body composition elements measured in Korean adult men and women based on the presence or absence of hypertension in each group. We then assessed the hypertension prediction accuracy when variables selected for their importance were used as features in a hypertension prediction model.

Before developing the model using machine learning techniques, we identified the variable importance factors, finding differences not only between men and women but also among the machine learning techniques themselves. Age was the common importance variable selected across six different machine learning techniques. The correlation between blood pressure and age has already been confirmed in various studies. In the study “On the interpretability of machine learning-based model for predicting hypertension”, age was presented as the highest importance feature for predicting the high risk of hypertension [28]. As age increases, blood vessels thicken and lose elasticity, and the decrease in heart pump function tends to increase systolic blood pressure [29]. This correlation is similarly mentioned in epidemiological studies like the Framingham Heart Study and other prior research that examined age-related changes in the risk of high blood pressure [30,31].

In Figure 3, to classify hypertension and non-hypertension, we identified the characteristics of the selected variables in both male and female groups. The common important variables for predicting hypertension in men were age, followed by SMM_WT, and then BFMp. In women, it was the opposite, with BFMp first, followed by SMM_WT. Prior research indicates that blood pressure is inversely associated with muscle mass and positively with fat mass [32]. Generally, men have a higher proportion of skeletal muscle mass and less fat mass compared to women. Besides this influence, the SLM and SMM of the recruited women in this study showed no statistical difference between the hypertension and non-hypertension groups; only the percentage divided by weight, SMM_WT, showed a difference, which seems to reflect the characteristics of these groups.

In Table 3, we identified the body composition variables with significant differences between the hypertension and non-hypertension groups for both men and women. In both groups, there was a statistically significant difference in the overall variable of body fat mass. However, for lean mass and muscle mass, the difference was observed only in the male group, with the hypertension group’s SLM, SMM, and SMM_WT being relatively lower than in the non-hypertension group. Overall, hypertension affects not only body fat mass but also muscle mass in men, whereas in women, body fat mass has a more significant impact than muscle mass.

In Table 4, examining the odds ratio results from the logistic regression analysis, it is evident that for both men and women, the hypertension group has approximately double the body fat mass variables compared to the non-hypertension group. Looking at the muscle variables, in men, the muscle mass in the hypertension group is about 0.33 to 0.67 times that of the non-hypertension group. This indicates that, for men, muscle mass is a more critical factor, suggesting that efforts to increase muscle mass, in addition to reducing body fat, are necessary for preventing hypertension. According to prior research, when the percentage of body fat exceeds 35%, it has a significant impact compared to when it is below 35%, with this impact being relatively higher in women than in men [33]. 

In Table 5, the performance metrics for the test set, commonly used as a classification performance indicator, are shown in both the men’s and women’s groups, where the E-net model demonstrated the best performance. 

Elastic-net regression is one of the regression methods developed to overcome the drawbacks of Ridge and Lasso regressions [34]. Elastic-net regression applies a combination of penalties and offers the advantage of performing both variable selection and model shrinkage simultaneously [35]. Additionally, the results of logistic regression can be quantitatively explained by using the odds ratio to interpret the weight of the variables between the two groups. 

Among the men’s group, the AUROC values for the five models, excluding E-net, were relatively high, ranging from 0.7 to 0.8. In the validation results based on the test set, NN, XGBoost, and K-NN showed the highest performance in that order. All three machine learning techniques, except for RF and SVM, showed no significant difference in the AUROC test values compared to E-net. When examining other metrics besides the AUROC, such as F1 score, precision, accuracy, sensitivity, and specificity, NN demonstrated the highest performance and exhibited a similar performance to E-net. The NN method has advantages in classifying complex patterns in data that cannot be explained by simple linear models. It also has advantages in handling high-dimensional data. However, its shortcomings include the potential for overfitting and the need for complex computations. The most significant disadvantage is the lack of explainability, as it is limited in providing reasons for its classifications.

In the women’s group, the AUROC for E-net was the highest, similar to the models in the men’s group. E-net also showed the best performance in sensitivity, while other metrics favored XGBoost. This is likely due to the influence of the selected variables. In the men’s group, three variables were selected by E-net, whereas in the women’s group, nine variables were selected. Having fewer selected variables offers advantages such as reduced curse of dimensionality, prevention of overfitting, increased computational efficiency, and reduced noise. Therefore, E-net’s best performance can be attributed to the smaller number of selected variables. Unlike the models in the men’s group, the women’s group models showed no statistically significant difference in AUROC compared to E-net. This suggests that when evaluating model performance based on AUROC, using various models can achieve similar classification performance. While model performance can be assessed using various criteria such as AUROC and other metrics, using classification techniques like E-net, which offer better explainability, may be advantageous for evaluating the weights or influence of each variable. 

Previous studies using body composition features to classify hypertension and non-hypertension groups have demonstrated the feasibility of logistic regression for this purpose. For instance, classification models using variables obtained from Bioelectrical Impedance Analysis (BIA) achieved classification accuracies between 83% and 90% [12]. Another study developed classification models incorporating general characteristics such as age, height, weight, and BMI alongside body composition variables, achieving over 70% classification accuracy [36]. When classification accuracy is ensured, logistic regression models can identify the weights of each variable, contributing to the classification and prediction of hypertension. 

The limitation of this study is the imbalance in the number of subjects between the hypertension and non-hypertension groups. Additionally, there is a need for more detailed investigations considering age-specific group characteristics to better understand the occurrence of hypertension across different age groups. Future research aims to address these limitations by including a larger sample of hypertension subjects, allowing for more detailed analysis and model development. In follow-up studies, we plan to compensate for limitations through more detailed analysis and model development by comparing more hypertension groups.

## 5. Conclusions

This study aims to develop a prediction model that can classify the presence or absence of hypertension in Korean adults aged 20 or older using various machine learning techniques. A common variable of interest across various machine learning techniques to investigate the association between individual body composition variables and hypertension was “age”. For men, the most important variables were skeletal muscle mass and body fat mass, and for women, they were BFM and SMM, in that order. Therefore, it can be concluded that for men, SMM is a more important factor, and efforts to increase muscle mass while reducing fat mass are important to prevent hypertension. Additionally, among the various classification techniques, E-net demonstrated the best performance when evaluating models based on the AUROC in both the men’s and women’s groups. It is important to choose and use a classification technique that aligns with the characteristics of the data in the group being classified, considering factors such as model explainability, the potential for overfitting, and the selected variables.

## Figures and Tables

**Figure 1 bioengineering-11-00921-f001:**
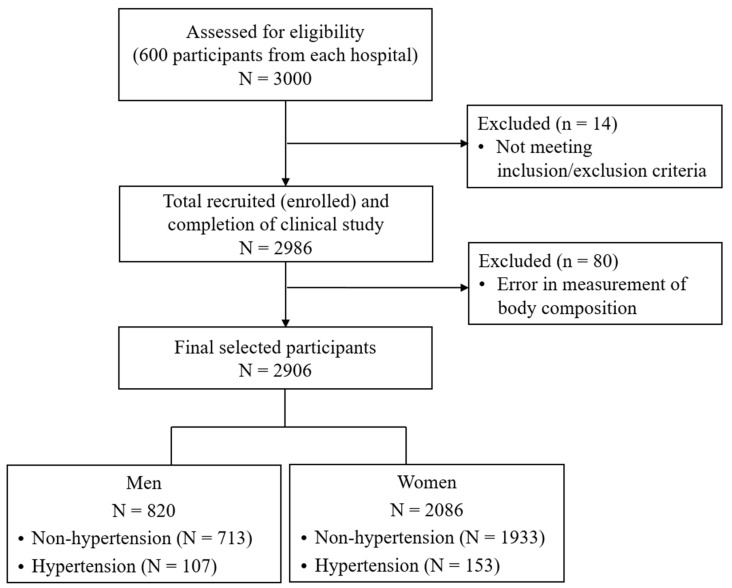
Flow chart of study sample selection.

**Figure 2 bioengineering-11-00921-f002:**
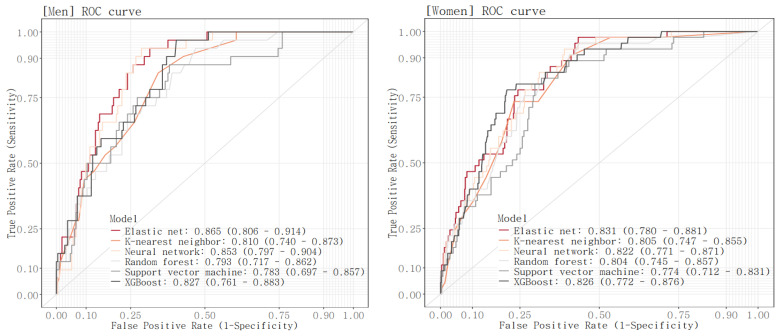
ROC curves of the six models.

**Figure 3 bioengineering-11-00921-f003:**
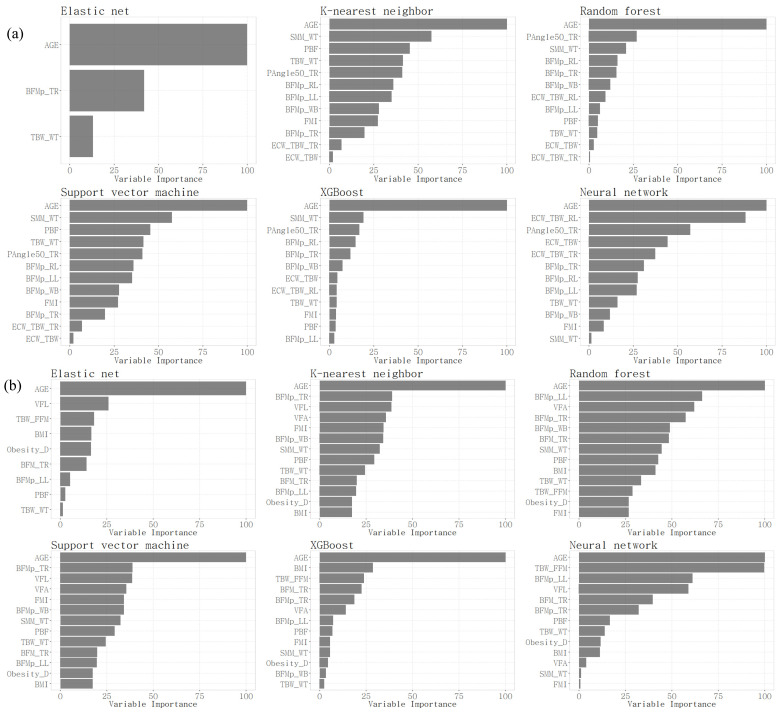
Relative importance of selected variables in men (**a**) and women (**b**).

**Table 1 bioengineering-11-00921-t001:** Description of body composition indicators.

Variables	Description
Body fat mass	
BFM	Body fat mass
PBF	Percentage of body fat mass to weight
FMI	Fat mass index
BFMp_WB †	Body fat mass (%) of whole body
BFM_RA	Body fat mass of right arm
BFMp_RA †	Body fat mass (%) of right arm
BFM_LA	Body fat mass of left arm
BFMp_LA †	Body fat mass (%) of left arm
BFM_TR	Body fat mass of trunk
BFMp_TR †	Body fat mass (%) of trunk
BFM_RL	Body fat mass of right leg
BFMp_RL †	Body fat mass (%) of right leg
BFM_LL	Body fat mass of left leg
BFMp_LL †	Body fat mass (%) of left leg
Lean mass	
SLM	Soft lean mass
SMM	Skeletal muscle mass
SMM_WT	Percentage of skeletal muscle mass to weight
Body water	
TBW	Total body water
ICW	Intracellular water
ECW	Extracellular water
TBW_WT	Percentage of total body water to weight
ECW_TBW	Proportion of extracellular water to total body water
ECW_TBW_RA	Proportion of extracellular water to total body water of right arm
ECW_TBW_LA	Proportion of extracellular water to total body water of left arm
ECW_TBW_TR	Proportion of extracellular water to total body water of trunk
ECW_TBW_RL	Proportion of extracellular water to total body water of right leg
ECW_TBW_LL	Proportion of extracellular water to total body water of left leg
Additional data	
VFL	Visceral fat level
VFA	Visceral fat area
Obesity_D	Proportion of weight to ideal weight
TBW_FFM	Proportion of total body water to fat-free mass
PAngle50_TR	50 kHz phase angle of trunk

† 100% refers to the appropriate amount of body fat mass at standard body weight.

**Table 2 bioengineering-11-00921-t002:** General characteristics and body composition indicators between the training set and test set.

Variables	Men			Women		
	Training Set	Test Set	*p*	Training Set	Test Set	*p*
Participants (n)	575	245		1462	624	
Hypertension			1			0.961
Yes	75 (13.04)	32 (13.06)		108 (7.39)	45 (7.21)	
No	500 (86.96)	213 (86.94)		1354 (92.61)	579 (92.79)	
Age (years)	44.27 ± 14.39	43.65 ± 14.57	0.574	46.62 ± 12.25	46.7 ± 12.45	0.887
Temperature (°C)	36.51 ± 0.33	36.51 ± 0.32	0.998	36.70 ± 0.32	36.71 ± 0.33	0.726
SBP (mmHg)	119.59 ± 14.07	120.09 ± 14.29	0.649	108.39 ± 17.15	109.24 ± 17.54	0.307
DBP (mmHg)	70.03 ± 10.14	70.17 ± 10.35	0.864	63.22 ± 10.73	64.02 ± 10.97	0.123
Pulse rate (bmp)	68.93 ± 10.04	67.77 ± 9.77	0.123	71.54 ± 10.31	70.89 ± 9.87	0.175
Height (cm)	172.20 ± 6.33	172.19 ± 6.16	0.983	159.40 ± 5.64	159.60 ± 5.62	0.450
Weight (kg)	74.47 ± 11.43	74.67 ± 10.61	0.804	58.74 ± 9.30	59.16 ± 9.35	0.340
BMI (kg/m^2^)	25.07 ± 3.22	25.16 ± 3.14	0.681	23.12 ± 3.51	23.24 ± 3.52	0.509
Body fat mass						
BFM	17.75 ± 6.43	18.02 ± 6.17	0.569	19.42 ± 6.51	19.56 ± 6.29	0.643
PBF	23.39 ± 5.87	23.73 ± 5.86	0.452	32.41 ± 6.31	32.45 ± 6.11	0.908
FMI	5.99 ± 2.14	6.09 ± 2.11	0.533	7.67 ± 2.61	7.71 ± 2.51	0.753
BFMp_WB	181.46 ± 64.69	184.44 ± 64.11	0.543	190.05 ± 82.71	191.23 ± 79.73	0.760
BFM_RA	1.05 ± 0.59	1.07 ± 0.56	0.677	1.37 ± 0.64	1.38 ± 0.59	0.715
BFMp_RA	178.93 ± 98.29	182.73 ± 96.08	0.607	153.51 ± 71.42	154.20 ± 66.08	0.830
BFM_LA	1.08 ± 0.59	1.10 ± 0.56	0.648	1.40 ± 0.64	1.40 ± 0.59	0.776
BFMp_LA	183.72 ± 98.44	187.49 ± 96.42	0.611	156.33 ± 71.81	156.68 ± 66.15	0.915
BFM_TR	9.11 ± 3.52	9.29 ± 3.43	0.495	9.42 ± 3.35	9.50 ± 3.29	0.601
BFMp_TR	220.63 ± 83.51	225.34 ± 83.85	0.461	188.17 ± 67.78	189.44 ± 66.41	0.691
BFM_RL	2.69 ± 0.86	2.72 ± 0.8	0.697	3.10 ± 0.93	3.12 ± 0.89	0.669
BFMp_RL	160.18 ± 50.85	161.64 ± 48.9	0.699	136.41 ± 41.79	136.89 ± 39.76	0.803
BFM_LL	2.67 ± 0.84	2.69 ± 0.8	0.729	3.09 ± 0.92	3.11 ± 0.88	0.717
BFMp_LL	158.72 ± 50.12	160.3 ± 48.77	0.675	135.88 ± 41.39	136.33 ± 39.40	0.814
Lean mass						
SLM	53.59 ± 6.93	53.53 ± 6.54	0.905	37.02 ± 4.16	37.28 ± 4.29	0.196
SMM	31.80 ± 4.47	31.74 ± 4.22	0.858	21.10 ± 2.64	21.27 ± 2.73	0.194
SMM_WT	42.90 ± 3.51	42.70 ± 3.51	0.438	36.24 ± 3.46	36.24 ± 3.33	0.998
Body water						
TBW	41.71 ± 5.36	41.66 ± 5.07	0.915	28.89 ± 3.24	29.1 ± 3.33	0.188
ICW	25.92 ± 3.43	25.87 ± 3.24	0.863	17.71 ± 2.03	17.84 ± 2.09	0.197
ECW	15.79 ± 1.96	15.79 ± 1.87	0.992	11.18 ± 1.23	11.26 ± 1.26	0.180
TBW_WT	56.35 ± 4.32	56.10 ± 4.30	0.447	49.68 ± 4.63	49.64 ± 4.49	0.880
ECW_TBW †	378.91 ± 7.56	379.29 ± 8.03	0.526	386.98 ± 6.02	386.95 ± 6.27	0.918
ECW_TBW_RA †	375.18 ± 4.3	375.39 ± 4.71	0.548	377.98 ± 3.63	378.26 ± 3.58	0.103
ECW_TBW_LA †	375.54 ± 4.55	375.59 ± 4.93	0.897	378.58 ± 3.67	378.69 ± 3.60	0.525
ECW_TBW_TR †	378.29 ± 7.46	378.65 ± 7.87	0.545	386.96 ± 5.86	386.94 ± 6.10	0.944
ECW_TBW_RL †	379.43 ± 9.38	380.06 ± 10.47	0.419	388.57 ± 7.73	388.53 ± 8.34	0.911
ECW_TBW_LL †	382.36 ± 9.74	382.61 ± 9.72	0.736	390.57 ± 7.56	390.38 ± 7.65	0.604
Additional data						
VFL	7.02 ± 2.96	7.22 ± 3.05	0.401	8.72 ± 3.66	8.78 ± 3.59	0.732
VFA	75.34 ± 29.97	77.43 ± 30.45	0.366	92.14 ± 37.16	92.68 ± 35.95	0.753
Obesity_D	113.94 ± 14.62	114.38 ± 14.24	0.688	110.12 ± 16.73	110.63 ± 16.76	0.525
TBW_FFM	73.55 ± 0.25	73.56 ± 0.25	0.687	73.5 ± 0.21	73.49 ± 0.23	0.278
PAngle50_TR	7.38 ± 0.91	7.34 ± 0.88	0.516	5.81 ± 0.52	5.79 ± 0.57	0.357

† The numbers represent values multiplied by 1000. Abbreviations: SBP, systolic blood pressure; DBP, diastolic blood pressure; BMI, body mass index; bmp, beats per minute; *p*, *p*-value. Continuous variables were summarized as mean ± standard deviation. *p*-values were obtained from independent two-sample t-tests for continuous variables and chi-square tests for categorical variables between the training and test sets by sex.

**Table 3 bioengineering-11-00921-t003:** Comparison of general characteristics and body composition indicators between hypertension and non-hypertension groups.

Variables	Men			Women		
	Non-Hypertension	Hypertension	*p*	Non-Hypertension	Hypertension	*p*
Participants (n)	500	75		1354	108	
General characteristics						
Age (years)	41.96 ± 13.42	59.68 ± 10.72	<0.001	45.76 ± 12.09	57.36 ± 8.59	<0.001
Temperature (°C)	36.51 ± 0.33	36.50 ± 0.34	0.700	36.70 ± 0.32	36.70 ± 0.31	0.952
SBP (mmHg)	118.52 ± 13.45	126.75 ± 16	<0.001	107.04 ± 16.58	125.20 ± 15.24	<0.001
DBP (mmHg)	69.50 ± 9.79	73.56 ± 11.71	0.005	62.51 ± 10.55	72.03 ± 9.08	<0.001
Pulse rate (bmp)	68.96 ± 10.10	68.72 ± 9.63	0.839	71.61 ± 10.22	70.58 ± 11.43	0.365
Height (cm)	172.63 ± 6.16	169.36 ± 6.76	<0.001	159.56 ± 5.61	157.41 ± 5.63	<0.001
Weight (kg)	74.29 ± 11.21	75.66 ± 12.83	0.383	58.34 ± 8.99	63.65 ± 11.56	<0.001
BMI (kg/m^2^)	24.88 ± 3.16	26.29 ± 3.38	0.001	22.92 ± 3.36	25.66 ± 4.33	<0.001
Body fat mass						
BFM	17.21 ± 6.31	21.37 ± 6.08	<0.001	19.06 ± 6.25	23.95 ± 7.92	<0.001
PBF	22.72 ± 5.75	27.89 ± 4.61	<0.001	32.06 ± 6.20	36.88 ± 5.97	<0.001
FMI	5.78 ± 2.08	7.43 ± 1.97	<0.001	7.51 ± 2.49	9.67 ± 3.19	<0.001
BFMp_WB	174.92 ± 62.93	225.05 ± 59.50	<0.001	184.98 ± 78.92	253.62 ± 101.20	<0.001
BFM_RA	1.00 ± 0.58	1.38 ± 0.58	<0.001	1.34 ± 0.60	1.81 ± 0.91	<0.001
BFMp_RA	169.31 ± 95.04	243.1 ± 96.04	<0.001	149.2 ± 66.58	207.52 ± 102.03	<0.001
BFM_LA	1.03 ± 0.58	1.42 ± 0.59	<0.001	1.36 ± 0.60	1.84 ± 0.90	<0.001
BFMp_LA	173.82 ± 95.16	249.72 ± 94.97	<0.001	152.03 ± 67.03	210.24 ± 102.11	<0.001
BFM_TR	8.83 ± 3.46	11.03 ± 3.33	<0.001	9.23 ± 3.24	11.79 ± 3.76	<0.001
BFMp_TR	212.46 ± 81.49	275.12 ± 76.53	<0.001	183.96 ± 65.24	240.94 ± 76.61	<0.001
BFM_RL	2.62 ± 0.84	3.18 ± 0.78	<0.001	3.06 ± 0.89	3.70 ± 1.18	<0.001
BFMp_RL	154.88 ± 49.50	195.51 ± 45.57	<0.001	134.01 ± 39.81	166.44 ± 53.14	<0.001
BFM_LL	2.60 ± 0.83	3.16 ± 0.77	<0.001	3.05 ± 0.89	3.68 ± 1.15	<0.001
BFMp_LL	153.49 ± 48.80	193.6 ± 44.82	<0.001	133.51 ± 39.49	165.57 ± 52.09	<0.001
Lean mass						
SLM	53.93 ± 6.76	51.3 ± 7.65	0.006	36.99 ± 4.12	37.43 ± 4.67	0.348
SMM	32.05 ± 4.36	30.15 ± 4.88	0.002	21.09 ± 2.62	21.27 ± 2.94	0.538
SMM_WT	43.34 ± 3.41	39.99 ± 2.73	<0.001	36.44 ± 3.40	33.76 ± 3.21	<0.001
Body water						
TBW	41.96 ± 5.23	39.99 ± 5.94	0.008	28.87 ± 3.20	29.23 ± 3.64	0.310
ICW	26.11 ± 3.35	24.65 ± 3.74	0.002	17.70 ± 2.01	17.84 ± 2.25	0.536
ECW	15.86 ± 1.91	15.34 ± 2.23	0.06	11.16 ± 1.22	11.39 ± 1.40	0.101
TBW_WT	56.83 ± 4.24	53.12 ± 3.39	<0.001	49.93 ± 4.56	46.47 ± 4.38	<0.001
ECW_TBW †	378.19 ± 7.30	383.71 ± 7.58	<0.001	386.77 ± 5.97	389.62 ± 6.07	<0.001
ECW_TBW_RA †	374.87 ± 4.14	377.28 ± 4.75	<0.001	377.92 ± 3.63	378.71 ± 3.53	0.027
ECW_TBW_LA †	375.24 ± 4.32	377.60 ± 5.47	0.001	378.54 ± 3.70	379.09 ± 3.14	0.086
ECW_TBW_TR †	377.56 ± 7.18	383.13 ± 7.48	<0.001	386.75 ± 5.81	389.52 ± 6.01	<0.001
ECW_TBW_RL †	378.59 ± 9.19	385.05 ± 8.74	<0.001	388.30 ± 7.65	392.01 ± 7.88	<0.001
ECW_TBW_LL †	381.48 ± 9.46	388.19 ± 9.67	<0.001	390.28 ± 7.47	394.28 ± 7.73	<0.001
Additional data						
VFL	6.75 ± 2.86	8.85 ± 2.98	<0.001	8.49 ± 3.54	11.55 ± 4.01	<0.001
VFA	72.56 ± 29.00	93.85 ± 29.93	<0.001	89.89 ± 35.84	120.32 ± 41.72	<0.001
Obesity_D	113.10 ± 14.34	119.51 ± 15.28	0.001	109.16 ± 16.00	122.19 ± 20.64	<0.001
TBW_FFM	73.53 ± 0.24	73.67 ± 0.22	<0.001	73.49 ± 0.21	73.63 ± 0.19	<0.001
PAngle50_TR	7.49 ± 0.87	6.67 ± 0.85	<0.001	5.83 ± 0.52	5.64 ± 0.47	<0.001

† The numbers represent values multiplied by 1000. Abbreviations: SBP, systolic blood pressure; DBP, diastolic blood pressure; BMI, body mass index; bmp, beats per minute; *p*, *p*-value. Continuous variables were summarized as mean ± standard deviation. *p* values were obtained from two-sample t-test between hypertension and non-hypertension groups by sex.

**Table 4 bioengineering-11-00921-t004:** Association of individual body composition indices with hypertension.

Variables	Men		Women	
	Crude OR (95% CI)	*p*	Crude OR (95% CI)	*p*
General characteristics				
Age (years)	4.32 (3.09, 6.06)	<0.001	3.36 (2.57, 4.4)	<0.001
Height (cm)	0.58 (0.45, 0.75)	<0.001	0.68 (0.56, 0.83)	<0.001
Weight (kg)	1.12 (0.89, 1.42)	0.333	1.58 (1.34, 1.85)	<0.001
BMI (kg/m^2^)	1.50 (1.20, 1.89)	0.001	1.82 (1.55, 2.14)	<0.001
Body fat mass				
BFM	1.77 (1.41, 2.22)	<0.001	1.78 (1.51, 2.09)	<0.001
PBF	2.70 (2.02, 3.61)	<0.001	2.20 (1.79, 2.71)	<0.001
FMI	2.03 (1.60, 2.58)	<0.001	1.89 (1.60, 2.22)	<0.001
BFMp_WB	2.03 (1.60, 2.58)	<0.001	1.89 (1.60, 2.22)	<0.001
BFM_RA	1.68 (1.35, 2.09)	<0.001	1.63 (1.40, 1.89)	<0.001
BFMp_RA	1.86 (1.48, 2.33)	<0.001	1.70 (1.46, 1.98)	<0.001
BFM_LA	1.71 (1.37, 2.13)	<0.001	1.63 (1.40, 1.89)	<0.001
BFMp_LA	1.90 (1.51, 2.39)	<0.001	1.70 (1.46, 1.97)	<0.001
BFM_TR	1.78 (1.41, 2.25)	<0.001	1.88 (1.58, 2.23)	<0.001
BFMp_TR	2.04 (1.60, 2.61)	<0.001	2.00 (1.68, 2.38)	<0.001
BFM_RL	1.79 (1.43, 2.25)	<0.001	1.69 (1.44, 1.98)	<0.001
BFMp_RL	2.07 (1.63, 2.63)	<0.001	1.79 (1.53, 2.11)	<0.001
BFM_LL	1.80 (1.44, 2.26)	<0.001	1.69 (1.44, 1.98)	<0.001
BFMp_LL	2.08 (1.64, 2.64)	<0.001	1.8 0(1.53, 2.11)	<0.001
Lean mass				
SLM	0.67 (0.51, 0.87)	0.002	1.11 (0.91, 1.34)	0.295
SMM	0.64 (0.49, 0.83)	0.001	1.07 (0.88, 1.30)	0.495
SMM_WT	0.33 (0.24, 0.44)	<0.001	0.44 (0.35, 0.54)	<0.001
Body water				
TBW	0.68 (0.52, 0.88)	0.003	1.12 (0.92, 1.35)	0.256
ICW	0.64 (0.49, 0.83)	0.001	1.07 (0.88, 1.3)	0.494
ECW	0.76 (0.59, 0.98)	0.034	1.2 (0.99, 1.44)	0.063
TBW_WT	0.38 (0.29, 0.51)	<0.001	0.46 (0.37, 0.57)	<0.001
ECW_TBW	2.08 (1.61, 2.69)	<0.001	1.62 (1.33, 1.99)	<0.001
ECW_TBW_RA	1.74 (1.36, 2.22)	<0.001	1.25 (1.02, 1.52)	0.029
ECW_TBW_LA	1.66 (1.3, 2.11)	<0.001	1.16 (0.95, 1.42)	0.133
ECW_TBW_TR	2.09 (1.62, 2.69)	<0.001	1.61 (1.32, 1.97)	<0.001
ECW_TBW_RL	2.01 (1.56, 2.59)	<0.001	1.64 (1.34, 2)	<0.001
ECW_TBW_LL	2.05 (1.58, 2.67)	<0.001	1.72 (1.41, 2.11)	<0.001
Additional data				
VFL	1.88 (1.5, 2.37)	<0.001	2.03 (1.7, 2.42)	<0.001
VFA	1.86 (1.48, 2.35)	<0.001	1.95 (1.65, 2.32)	<0.001
Obesity_D	1.5 (1.19, 1.9)	0.001	1.82 (1.55, 2.14)	<0.001
TBW_FFM	1.82 (1.4, 2.37)	<0.001	1.9 (1.56, 2.32)	<0.001
PAngle50_TR	0.36 (0.27, 0.49)	<0.001	0.69 (0.56, 0.84)	<0.001

Abbreviations: SBP, systolic blood pressure; DBP, diastolic blood pressure; *p*, *p*-value; OR, odds ratio; CI, confidence interval. *p* values were obtained from simple logistic regression analyses for individual body composition variables by sex.

**Table 5 bioengineering-11-00921-t005:** Performance of the six models.

Men’ Models	Training Set AUROC	Test Set AUROC (95% CI)	AUROC Test	*p*	Kappa (95% CI)	F1 Score (95% CI)	Precision (95% CI)	Accuracy (95% CI)	Sensitivity (95% CI)	Specificity (95% CI)
E-net	0.874	0.865 (0.806, 0.914)			0.335 (0.228, 0.435)	0.466 (0.348, 0.563)	0.310 (0.215, 0.400)	0.718 (0.657, 0.771)	0.941 (0.833, 1.000)	0.686 (0.620, 0.742)
K-NN	0.876	0.81 (0.74, 0.873)	E-net vs. K-NN	0.075	0.264 (0.168, 0.362)	0.409 (0.301, 0.507)	0.271 (0.186, 0.355)	0.682 (0.620, 0.739)	0.848 (0.707, 0.966)	0.659 (0.593, 0.718)
RF	1	0.793 (0.717, 0.862)	E-net vs. RF	0.045	0.203 (0.132, 0.286)	0.372 (0.275, 0.459)	0.231 (0.162, 0.303)	0.584 (0.522, 0.645)	0.941 (0.840, 1.000)	0.530 (0.462, 0.600)
SVM	0.959	0.783 (0.697, 0.857)	E-net vs. SVM	0.018	0.243 (0.154, 0.337)	0.395 (0.292, 0.493)	0.256 (0.177, 0.340)	0.653 (0.592, 0.710)	0.879 (0.750, 0.971)	0.619 (0.553, 0.682)
XGBoost	0.956	0.827 (0.761, 0.883)	E-net vs. XGBoost	0.433	0.261 (0.185, 0.354)	0.413 (0.312, 0.514)	0.263 (0.188, 0.345)	0.645 (0.584, 0.706)	0.970 (0.893, 1.000)	0.597 (0.531, 0.659)
NN	0.9	0.853 (0.797, 0.904)	E-net vs. NN	0.757	0.365 (0.257, 0.469)	0.489 (0.37, 0.586)	0.330 (0.232, 0.424)	0.743 (0.69, 0.796)	0.941 (0.833, 1.000)	0.714 (0.653, 0.771)
**Women’s** **Models**	**Training Set** **AUROC**	**Test Set** **AUROC** **(95% CI)**	**AUROC Test**	** *p* **	**Kappa** **(95% CI)**	**F1 Score** **(95% CI)**	**Precision** **(95% CI)**	**Accuracy** **(95% CI)**	**Sensitivity** **(95% CI)**	**Specificity** **(95% CI)**
E-net	0.833	0.831 (0.78, 0.881)			0.153 (0.111, 0.201)	0.259 (0.197, 0.322)	0.149 (0.110, 0.192)	0.597 (0.558, 0.636)	0.979 (0.927, 1.000)	0.568 (0.527, 0.607)
K-NN	0.849	0.805 (0.747, 0.855)	E-net vs. K-NN	0.399	0.224 (0.142, 0.299)	0.312 (0.225, 0.391)	0.198 (0.136, 0.260)	0.768 (0.732, 0.798)	0.735 (0.600, 0.857)	0.770 (0.733, 0.802)
RF	1	0.804 (0.745, 0.857)	E-net vs. RF	0.435	0.145 (0.104, 0.190)	0.251 (0.191, 0.312)	0.144 (0.107, 0.187)	0.591 (0.553, 0.628)	0.957 (0.889, 1.000)	0.563 (0.524, 0.603)
SVM	1	0.774 (0.712, 0.831)	E-net vs. SVM	0.117	0.179 (0.122, 0.242)	0.276 (0.209, 0.349)	0.166 (0.12, 0.219)	0.692 (0.655, 0.728)	0.829 (0.708, 0.932)	0.683 (0.644, 0.720)
XGBoost	0.915	0.826 (0.772, 0.876)	E-net vs. XGBoost	0.910	0.265 (0.186, 0.345)	0.347 (0.262, 0.430)	0.223 (0.162, 0.291)	0.790 (0.758, 0.821)	0.780 (0.653, 0.896)	0.791 (0.758, 0.824)
NN	0.846	0.822 (0.771, 0.871)	E-net vs. NN	0.780	0.165 (0.117, 0.220)	0.268 (0.206, 0.334)	0.157 (0.116, 0.203)	0.633 (0.595, 0.671)	0.938 (0.850, 1.000)	0.611 (0.569, 0.649)

Abbreviations: CI, confidence interval; E-net, elastic net; K-NN, k-nearest neighbor; RF, random forest; SVM, support vector machine; XGBoosting, extreme gradient boosting; NN, neural network; CI, confidence interval; *p, p*-value. The 95% CIs were obtained from 2000 bootstrap repetitions. The optimal threshold was determined using Youden’s index.

## Data Availability

The data presented in this study are available upon request from the corresponding author. These data are not publicly available because of privacy concerns.

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
