# Peer review of "Machine Learning Approach for Predicting Hypertension Based on Body Composition in South Korean Adults"

_bioengineering, 2024, doi:10.3390/bioengineering11090921_

Round 1
Reviewer 1 Report
Comments and Suggestions for Authors
Authors used Elastic-net for predicting hypertension in the Korean adults. But the manuscript suffers in the following aspects and mentioned below. I am not convinced about the novelty of the manuscript. The novelty of the paper needs to be justified and clearly defined. It includes a clear difference between the available literature and previous works. The authors are asked to provide the limitations of the previous correlated works and then link those limitations to the current ideas and contributions of the current work.
Interpretability of the model is lagging
Advantages and shortcoming of the network are not written
Why not lightweight CNN and RNN?
Can the code and data be shared for reproducing the results?
Figure 1 need to be properly arranged.
Table 2 needs to be properly represented.
Fig.5 needs to be properly represented and explained.
Can the code and data be shared for reproducing the results.
Comments on the Quality of English Languageneeds improvement.
Author Response
Response to Reviewer 1 Comments
Thank you for your review comments on this manuscript. I think it helped a lot to improve the completeness of this manuscript by your review comment. I actively responded to the lack of content given by the reviewer and tried to apply it to the manuscript. Once again, thank you for reviewing this manuscript, and please check the following for responses to comments.
Point 1: Interpretability of the model is lagging
Response 1: I agree with the your comments. It has been added to lines 358 to 383 of the Discussion to increase the interpretability of the model's performance. In each model for men and women, we added information about the highest performing technique and explanations of other techniques, as well as differences in selection variables as reasons for high performance. thank you.
Point 2: Advantages and shortcoming of the network are not written
Response 2: I agree with the your comments. The advantages and disadvantages of NN have been added in lines 364-369. The advantage of NN, high-dimensional processing ability, and the disadvantage, lack of explanatory power and possibility of overfitting, were mentioned. thank you.
Point 3: Why not lightweight CNN and RNN?
Response 3: I agree with the your comments. As you may already know, there are many reasons for using machine learning rather than deep learning such as CNN or RNN. First of all, deep learning techniques are difficult to explain due to the complexity of the generated model. Machine learning, on the other hand, provides simple and interpretable models, which has the advantage of understanding and explaining the decisions of the generated models. Since the selection of an appropriate technique varies depending on the specific situation and data, machine learning was used for this data. We thought that it would be difficult to use CNN, which specializes in image data, or RNN, which specializes in time series data, for this data. However, as you suggested, if there is a chance in the future, we will post-process this data and classify it to form a data set suitable for CNN or RNN. Please understand this. thank you
Point 4: Can the code and data be shared for reproducing the results?
Response 4: The code used in this analysis is from Caret's R package. This is presented in reference 14-16, and the code can be used by selecting available models at https://topepo.github.io/caret/available-models.html.
Our research team created a model using this code after simple post-processing and purification of the data. In addition, in the case of clinical data used for analysis, there are security issues that make it difficult to export it outside of the paper and conduct research due to IRB principles. Thank you for your understanding.
Point 5: Figure 1 need to be properly arranged.
Response 5: According to reviewer’s comments, we have rearranged Figure 1. thank you
Point 6: Table 2 needs to be properly represented.
Response 6: According to reviewer’s comments, we have represented Table 2. thank you
Point 7: Fig.5 needs to be properly represented and explained.
Response 7: An explanation of the selected variable has been added along with the model's performance in lines 323 to 332. Additionally, added at lines 358 to 383 of the Discussion. Thank you.
Point 8: Can the code and data be shared for reproducing the results.
Response 8: I answered with the same response as Point 4. Please understand it. Thank you.

Reviewer 2 Report
Comments and Suggestions for Authors
I appreciate the authors for presenting this research article, which emphasizes that hypertension affects not only body fat mass (BFM) but also skeletal muscle mass (SMM) in men, whereas in women, BFM has a more significant impact than SMM. My comments are as follows:
-
The authors mentioned that the Elastic-net technique demonstrated the highest classification accuracy. However, in Table 5, the Elastic-net technique showed no significant difference compared to K-NN, XGBoost, and NN in men, and K-NN, RF, SVM, XGBoost, and NN in women. Please clarify this point.
-
In Figure 4, why is SMM not the leading feature in terms of importance in the Elastic-net technique for men?
-
In Figure 4, please provide a brief discussion on why VEL is the leading feature in terms of importance in the Elastic-net technique for women.
Author Response
Response to Reviewer 2 Comments
Thank you for your review comments on this manuscript. I think it helped a lot to improve the completeness of this manuscript by your review comment. I actively responded to the lack of content given by the reviewer and tried to apply it to the manuscript. Once again, thank you for reviewing this manuscript, and please check the following for responses to comments.
Point 1: The authors mentioned that the Elastic-net technique demonstrated the highest classification accuracy. However, in Table 5, the Elastic-net technique showed no significant difference compared to K-NN, XGBoost, and NN in men, and K-NN, RF, SVM, XGBoost, and NN in women. Please clarify this point.
Response 1: I agree with the your comments. It has been added to lines 358 to 383 of the Discussion to increase the interpretability of the model's performance. In each model for men and women, we added information about the highest performing technique and explanations of other techniques, as well as differences in selection variables as reasons for high performance. thank you.
Point 2: In Figure 3, why is SMM not the leading feature in terms of importance in the Elastic-net technique for men?
Response 2: I agree with the your comments. In Table 3, the results of the T-test showed a difference by comparing the means of non-hypertension and hypertension within the men group, but there was no difference in the women group. Statistical comparison shows that SMM was an important variable, but was not selected as the selected variable of E-net in the development of the classification model. However, in XGBoost, which showed similar model performance, it was selected as the variable with the second highest importance. This is thought to reflect the characteristics of the models. I also think the cause of the linearity of the data. It is difficult to confirm the cause in detail in this study. We will investigate the cause in more detail in future research. Please understand. thank you
Point 3: In Figure 4, please provide a brief discussion on why VFL is the leading feature in terms of importance in the Elastic-net technique for women.
Response 3: The mention of VFL being selected as an important variable in the women’s classification model is addressed in lines 323-332. Thank you.

Round 2
Reviewer 1 Report
Comments and Suggestions for Authors
Authors used Elastic-net technique for demonstrating better classification accuracy. Explain about the network more detail with the no of computations. Also compare the no of computations with other networks.
But I am not convinced about the novelty of the manuscript. The novelty of the paper need to be explained properly.
Authors have used the many parametric values in Table 2. and Table 3. It needs to be represented properly.
Organization needs to be improved.
Represent Fig.3 and Fig.4 in a compact way.
Comments on the Quality of English LanguageNeeds improvement.
Author Response
Thank you for your additional review comments. Please understand that we have made 2nd round revisions as much as possible based on your opinions.
Point 1: Authors used Elastic-net technique for demonstrating better classification accuracy. Explain about the network more detail with the no of computations. Also compare the no of computations with other networks.
Response 1: Additionally, to compare the performance of Elastic-net and other techniques, we checked the statistical difference in AUROC values. The reason for this was discussed in lines 361 to 370 of the discussion. There are various methods for computational comparison absolute performance with other techniques, but in this study, we confirmed whether the AUROC value of each technique is statistically different from that of Elatic-net. I think the ways in which absolute comparisons can be made are limited. However, I think that an computational comparison is possible with the test-set results of models created using the same data. Please understand this.
Point 2: But I am not convinced about the novelty of the manuscript. The novelty of the paper need to be explained properly.
Response 2: This study is not a novel study aimed at developing new techniques and presenting machine learning techniques through new methods. However, the main purpose is to present the performance and method of classifying hypertension using body composition data measured in Korean adults. This novelty is comprehensively presented in lines 403-410 in the conclusion. Looking at previous studies, there are not many similar studies on the classification of hypertension in Koreans. Additionally, studies that classify men and women, present results, and identify the effected variables are rare. I think the “novelty” is talking about varies depending on perspective and direction. We ask that you acknowledge the novelty of this study, which was conducted on Koreans. Please understand this.
Point 3 : Authors have used the many parametric values in Table 2. and Table 3. It needs to be represented properly.
Response 3: The total value column in Table 2 has been deleted. We tried to delete unnecessary variables, but since they were measured values, we thought it would be appropriate to present them all, so we did not delete any variables. If the format of the table suitable for the journal's format is confirmed for future publication, the journal's editor and proofreading will modify it to fit the template and present it.
Point 4 : Organization needs to be improved.
Response 4: The detailed title of Method 2.1, 2.2, 2.3 has been modified. And 3.1 title of Result has been modified to be more concise. The organization of the content was presented in accordance with the research process. thank you
Point 5 : Represent Fig.3 and Fig.4 in a compact way.
Response 5: Figures 3 and 4 have been combined into one figure 3. We thought it would be better to present the contribution of each variable visually as a graph, so we presented it this way. We ask for your broad understanding.

Reviewer 2 Report
Comments and Suggestions for Authors
The authors have replied my comments item-by-item. I have no more comments. Accept is my final decision.
Author Response
Thank you for your review comments.
Point 1: The authors have replied my comments item-by-item. I have no more comments. Accept is my final decision.
Response 1: Your comments were very helpful in improving the quality of this manuscript. thank you.
